# Stilbene Content and Expression of Stilbene Synthase Genes in Korean Pine *Pinus koraiensis* Siebold & Zucc

Andrey R. Suprun *[ID], Alexandra S. Dubrovina [ID], Valeria P. Grigorchuk and Konstantin V. Kiselev *

Laboratory of Biotechnology, Federal Scientific Center of the East Asia Terrestrial Biodiversity, Far Eastern Branch of the Russian Academy of Sciences, Vladivostok 690022, Russia
* Correspondence: suprun@biosoil.ru (A.R.S.); kiselev@biosoil.ru (K.V.K.); Tel.: +8-423-2310410 (K.V.K.); Fax: +8-4232-310193 (K.V.K.)

**Abstract:** Stilbenes are a large group of plant phenolic compounds that have a wide range of biologically active properties, such as antioxidant, immunomodulatory, anti-inflammatory, and anti-angiogenic effects. In plants, stilbenes are involved in the defense against environmental stresses, including fungal infections and insect attacks. The biosynthesis of stilbenes is well described for those plant species where resveratrol and its derivatives are the predominant stilbenes. However, there is little information on stilbene biosynthesis in the Pinaceae family, although the highest content of stilbenes was found in plants of this family. In this study, seasonal variations in stilbene compositions and contents in different parts of *Pinus koraiensis* was described (needles, bark, wood, young branches, and strobiles). HPLC-ESI-MS analysis showed the presence of seven stilbenes in *P. koraiensis*: t-astringin; t-piceid; cis-piceid; t-isorapontin; t-pinostilbenoside; t-resveratrol; and t-pinostilbene. Glycosylated and methylated forms of stilbenes, such as t-astringin, t-piceid, and t-pinostilbenoside, prevailed over other stilbenoids. The highest content of stilbenes was detected in the bark collected in spring and winter (up to 54.8 mg/g dry weight). The complete protein-coding sequences of three stilbene synthase genes, *PkSTS1*, *PkSTS2*, and *PkSTS3*, were obtained from the RNA isolated from the *P. koraiensis* needles. The expression of the *PkSTS1*, *PkSTS2*, and *PkSTS3* genes was analyzed using real-time PCR and frequency analysis of cloned RT-PCR products in the needles of *P. koraiensis* collected in different seasons. Thus, we first analyzed stilbene biosynthesis in the different organs of pine *P. koraiensis* and *PkSTS* expression depending on the year seasons.

**Keywords:** *Pinus koraiensis*; pinaceae; astringin; isorhapontin; pinostilben; piceid; stilbenoid glucoside; *STS*; resveratrol; pinostilbenoside



## 1. Introduction

Korean pine (*Pinus koraiensis* Siebold & Zucc.) belongs to the Pinaceae family and is distributed in the Amur and Primorsky regions of Russia, in the north of Korea and northeast China, as well as in central Japan [1]. Korean pine is highly valued in construction due to the high quality of the wood [2]. The edible part of the seeds is called the pine nut. Pine nuts are edible and represent a rich source of protein, fiber, and unsaturated fatty acids [3].

Plant secondary metabolites are a large and diverse group of compounds that includes tens of thousands of substances, subdivided into several main classes: alkaloids, terpenoids or isoprenoids, and phenolic compounds [4]. Stilbenes are a large group of phenolic compounds found in plants belonging to various families [5], such as Vitaceae, e.g., grape [6]; Fabaceae, e.g., peanut [7,8]; Polygonaceae, e.g., knotweed, rhubarb [9,10]; Gnetaceae, e.g., gnetum [11]; Poaceae, e.g., sorghum [12]; and Ericaceae, e.g., bilberry and blueberries [13,14].

Stilbenes have valuable biologically active properties and can be used as food additives to increase the value of food, medicine [15], cosmetology [16], and agriculture [17]. Overall,

stilbene shows promise as a natural compound with potential health benefits [18], although more research is needed to fully understand its mechanisms of action and potential uses. Stilbenes are involved in constitutive and inducible plant defense responses and have also attracted intense interest due to their protective properties against plant fungal pathogens, nematodes, and herbivores [19–21]. They can also reduce oxidative stress by increasing plant defenses against abiotic stresses such as drought, low temperatures, or high light [22].

Species of the family Pinaceae synthesize a wide range of stilbenes such as pinosylvin, pinosylvin monomethyl ether, pinosylvin dimethyl ether, piceid, and resveratrol (Figure 1 and Table S1). It is interesting to note that different species of the Pinaceae family are characterized by different stilbene composition. For example, the genus *Pinus* is characterized by the presence of a large amount of pinostilbene and its glycosylated form or pinosylvin and methyl-pinosylvin (Figure 1 and Table S1) [23–30]. On the contrary, species of the genus *Picea* are characterized by the presence of high amounts of *t*-astringin and *t*-isorapontin (Figure 1 and Table S2) [31,32].

| Stilbene | Molecular Formula | Molar mass | R3 | R5 | R3' | R4' |
|---|---|---|---|---|---|---|
| t-resveratrol | $C_{14}H_{12}O_3$ | 228.25 | OH | OH | H | OH |
| t-piceid (polydatin) | $C_{20}H_{22}O_8$ | 390.4 | OH | OGlu | H | OH |
| t-piceatannol | $C_{14}H_{12}O_4$ | 244.24 | OH | OH | OH | OH |
| t-astringin | $C_{20}H_{22}O_9$ | 406.4 | OGlu | OH | OH | OH |
| t-pinosylvin | $C_{14}H_{12}O_2$ | 212.24 | OH | OH | H | H |
| t-pinosylvin monomethyl ether | $C_{15}H_{14}O_2$ | 226.27 | OCH3 | OH | H | H |
| t-pinosylvin dimethyl ether | $C_{16}H_{16}O_2$ | 240.3 | OCH3 | OCH3 | H | H |
| t-dihydropinosylvin monomethyl ether | $C_{15}H_{16}O_2$ | 228.29 | OCH3 | OH | H | H |
| t-pinostilben | $C_{15}H_{14}O_3$ | 242.27 | OCH3 | OH | H | OH |
| t-pinostilbenoside | $C_{21}H_{24}O_8$ | 404.4 | OCH3 | OH | H | OGlu |
| t-isorhapontigenin | $C_{15}H_{14}O_4$ | 258.27 | OH | OH | OCH3 | OH |
| t-isorhapontin | $C_{21}H_{24}O_9$ | 420.4 | OGlu | OH | OCH3 | OH |

**Figure 1.** Chemical structures of stilbenes of different Pinaceae species. OGlu—O-bD-glucopyranoside.

The biosynthesis of stilbenes in plants proceeds through the phenylpropanoid pathway [33]. Stilbene synthase (STS, EC 2.3.1.95) condenses three molecules of malonyl-CoA with one molecule of coumaryl-CoA to form resveratrol. Furthermore, resveratrol undergoes successive modifications, such as methylation by resveratrol-O-methyltransferase with the formation of pterostilbene [6]; glycosylation by glucosyltransferase to form piceid [34]; or oligomers, which include dimers, trimers, and tetramers, via the oxidation of resveratrol by polyphenol oxidase [19].

In plant cells, stilbenes exist as a mixture of *cis*- and *trans*-isomers, and the balance of these forms is usually strongly shifted towards the accumulation of a thermodynamically more stable *trans*-form [35,36]. The formation of the *cis*-form of stilbenes occurs under the influence of UV radiation from *trans*-stilbenes (*t*-stilbenes) [35]. In recent years, much has become known about the biosynthesis of stilbenes in representatives of the genus *Picea* [5,19]. Two genes encoding *STS* enzymes (*STS1* and *STS2*) from three spruce species, *Picea abies*, *Picea glauca*, and *Picea sitchensis*, have been described [37]. In addition, two full-length *STS* genes and two *STS* pseudogenes of white spruce *Picea glauca* have been described. [38]. Four transcripts encoding STS enzymes from the Ajan spruce *Picea jezoensis* have been isolated and sequenced [39]. It is known that the pine *Pinus sylvestris* contains

a stilbene synthase (pinosylvin synthase) gene, which is involved in the formation of pinosylvin, but not resveratrol [40].

In this study, we determined for the first time the composition and content of stilbenes in various parts of *P. koraiensis* trees in different seasons of the year and successfully cloned all expressed *STS* genes in *P. koraiensis* to study seasonal gene expression. The data obtained indicate that the highest amounts of stilbenes (53 mg/g of dry weight) accumulate in winter in the bark of *P. koraiensis*.

## 2. Materials and Methods

### 2.1. Plant Material

Samples of the pine *Pinus koraiensis*, namely, needles, branches, bark, and wood, were collected in winter, spring, summer, and autumn from three trees in the Far Eastern Federal District of Russia (longitude 43.8715338387169 and latitude 132.79473502372) in 2022. Additionally, in the autumn, pine strobiles were collected for analysis, which were divided into scales, seed shells, and pine nuts (the edible part). We collected the needles, branch, bark, and wood of 20–25-year-old trees (2–2.5 m from the soil). Within two hours after sample collection, the samples were dried at a temperature of 60 °C for 24 h and ground using an analytical mill IKA A11 basic (Koln, Germany). The crushed tissue was sieved to select particles ranging in size from 0.2 to 1 mm.

### 2.2. Stilbene Identification and Quantification by HPLC-DAD-ESI-MS

Samples for HPLC-DAD-ESI-MS analysis were prepared using 100 mg of dried and crushed tissues that were extracted with 3 mL of 96% ethanol at 60 °C for 2 h in the dark. After extraction, we purified samples with Discovery® DSC-18 SPE Tube bed wt. 50 mg, volume 1 mL (Supelco, Bellefonte, PA, USA). Stilbene content in all tissue types were analyzed three times for each sample. The targeted HPLC–electrospray ionization tandem mass spectrometry (HPLC-DAD-ESI-MS) of all stilbenes was fulfilled using a HPLC LC-20AD system (Shimadzu, Japan) connected with a LCMS-IT-TOF tandem mass-spectrometer (Shimadzu, Kyoto, Japan) [31,35]. The content of stilbenes was determined by the external standard method as described [35].

The chromatographic division was executed on C18 column Shim-pack GIST series (150 mm length, 2.1 mm internal diameter, 3 μm part size, Shimadzu, Japan). The column was kept at a constant temperature of 40 °C. The mobile phase consisted of A (0.2% aqueous acetic acid) and B (0.2% acetic acid in acetonitrile). The flow rate was constant and amounted to 0.2 mL/min. For the analysis of stilbenes, the following gradient program was used: 0.01 min 0% of B; 35 min 40% of B; 40 min 50% of B; 50 min 100% of B; 65 min 100% of B; 65.01 min 0% of B; 65.01 min 0% of B; and 80 min 0% of B. The volume of the injection sample was 2 μL.

The set of analytical standards comprised t-piceid and t-pinostilbene acquired from Sigma-Aldrich (St. Louis, MO, USA), t-astringin and t-isorhapontin obtained from Polyphenols (Sandnes, Norway), and t-resveratrol purchased from TCI (Tokyo Chemical Industry UK Ltd., Oxford, UK). Commercially available cis- forms of stilbenes were not obtainable. To determine and measure the cis- isomers of stilbenes, the corresponding standards were obtained from the t-standard, as described before [22]. Briefly, 1 mL of t-stilbene standard at 1000 ppm was exposed to sunlight for 3 h. Almost 80% isomerization was obtained after 3 h of exposure. The levels of the t- and cis-induced versions were calculated by measuring the high concentration of signals of quasi-molecular ions of the t- and cis- isomers relative to the use of extracted ion chromatograms by the method of obtaining HPLC-DAD-ESI-MS obtained from the calculations of ions.

All identified stilbenes of the extracts were identified based on DAD UV spectra, mass spectrum data, and chromatographic separation with reference to the values of the analytical standards [35]. UV spectra were recorded in the 190–400 nm diapason, and chromatograms for quantification were acquired at 310 nm. The amounts of each component were determined by using external standard method as described [35].

*2.3. Isolation, Cloning, and Sequencing of PkSTS Genes of P. koraiensis*

For the expression analysis of the *PkSTS* genes, we isolated RNA from the fresh needles of three 20–25-year-old pine *P. koraiensis*. To isolate total RNA from needles, an extraction method based on the use of hexadecyltrimethylammonium bromide (CTAB) was used [41,42]. Complementary DNAs (cDNA) were obtained using the kit ThermoScript reverse transcriptase (Invitrogen, Life Technologies, Carlsbad, CA, USA) according to the manufacturer's recommendations, as described previously [43]. Full length cDNAs of *P. koraiensis PkSTS1* (GenBank acc. no. OQ621419), *PkSTS2* (GenBank acc. no. OQ621420), and *PkSTS3* (GenBank acc. no. OQ621421) were amplified based on the known mRNA sequences of *P. densiflora*, *P. strobus*, and *P. massoniana*.

The primers 5′ATG TCTGTAGGAATGGGCG and 5′TTAAGGGAAAGGAATGCTC A were used to recover the full cDNA of the *PkSTSs* genes. Amplifications were performed using Tersus polymerase (Evrogen, Moscow, Russia) according to the manufacturer's recommendations. The target amplicons were isolated from the gel using kit Cleanup Mini (Evrogen, Moscow, Russia), cloned to pJET1.2/blunt (Thermo Fissher Scientific, Vilnius, Lithuania) and sequenced using a Big Dye Terminator Cycle Sequencing Kit (Perkin–Elmer Biosystems, Forster City, CA, USA) following the manufacturer's protocol and recommendations. After purification with ethanol, the sequences were identified on an ABI 3130 Genetic Analyser (Perkin–Elmer Biosystems). Next, using frequency analysis of cloned RT-PCR products [44], the occurrence of STS genes clones was calculated.

*2.4. Real-Time Quantitative Reverse Transcription PCR (qRT-PCR)*

The qRT-PCRs were amplified in the presence of EvaGreen fluorescent dye (Biotium, Fremont, CA, USA) using a Real-time PCR Kit (Evrogen, Moscow, Russia) in accordance with the manufacturer's protocol. For amplification, we used a DTprime unit with function of detection of results in real time (DNA Technology, Russia) as described [39,45]. Expression was calculated by the $2^{-\Delta\Delta CT}$ method [46]. Table S3 contains the primers utilized in our study.

Two housekeeping genes of pine (Actin (GenBank acc. no. MN172177) and GAPDH (GenBank acc. no. KM496531)) were used as controls to normalize variance in the quality and the amount of cDNA used in each qRT-PCR experiment. qRT-PCR data shown were obtained from at least three experiments and are averages of 3 technical replicates for each experiment.

*2.5. Statistical Analysis*

The data that was presented consisted of mean values with a standard error of the mean (SEM). In order to analyze the data collected from the experiments, a one-way analysis of variance (ANOVA) was conducted with Tukey's pairwise comparison test at $p = 0.05$. Data on the content of stilbenes were collected from three plants, with two repeated measurements taken for each.

## 3. Results

*3.1. Stilbene Identification and Quantification*

Seasonal fluctuations had a noticeable impact on the stilbene levels found in the *P. jezoensis* spruce needles, as evidenced by research [35]. Therefore, we analyzed stilbene composition and content in the different parts of the pine *P. koraiensis* collected at different year seasons to assess what season was the most appropriate time for sample collection and stilbene analysis. The samples of *P. koraiensis* were collected and extracted in spring, summer, autumn, and winter.

Using the HPLC-ESI-MS method, we detected seven stilbenoid compounds in *P. koraiensis*: t-astringin; t-piceid; cis-piceid; t-isorhapontin; t-resveratrol; t-pinostilbene; and t-pinostilbenoside (Figure 2). It is possible that other stilbenes were present in the samples of *P. koraiensis* but in trace amounts (less than 0.001 mg/g DW). The detection of other stilbenes may require changes in the tissue extraction protocol or an increase in the

sensitivity of the HPLC-ESI-MS method. In this work, we studied the main stilbenes that were found in the tissues of *P. koraiensis*, and, therefore, we analyzed the content of only the above compounds.

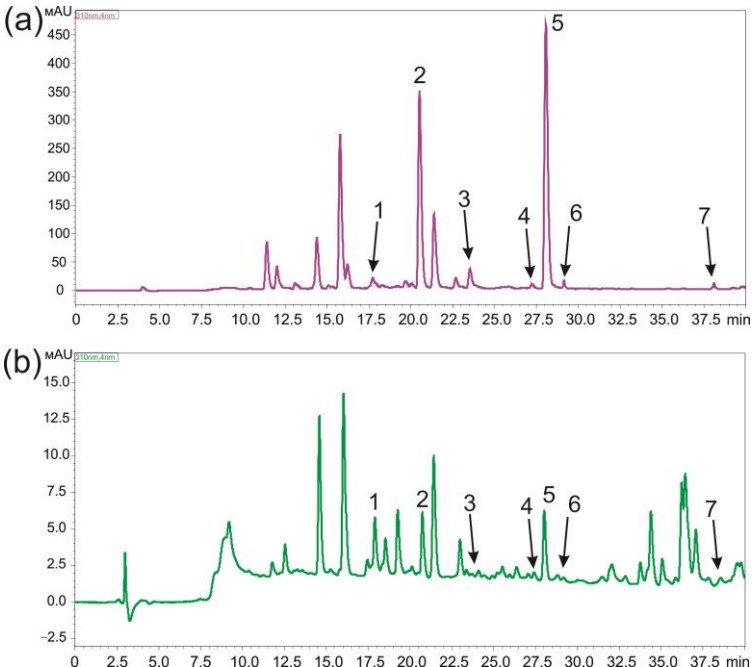

**Figure 2.** Comparison of HPLC-DAD-ESI-MS chromatographic profiles for the extracts of pine *Pinus koraiensis* bark (**a**) and needles (**b**) recorded at 310 nm. *P. koraiensis* bark and needles were collected in winter. *T*-astringin (1), *t*-piceid (2), *cis*-piceid (3), *t*-isorhapontin (4), *t*-pinostilbenoside (5), *t*-resveratrol (6), *t*-pinostilben (7).

The results obtained showed that the pinus bark contained significantly higher total levels of stilbenes in comparison with the needles, branches, and wood of pine *P. koraiensis* (Tables 1–4). The highest content of stilbenes in the pine bark was detected in the winter period: 53.8 mg/g DW (Table 1), which significantly decreased in spring (36.9 mg/g DW), summer (44.5 mg/g DW), and autumn (35.8 mg/g DW). The main stilbenes in all pine *P. koraiensis* probes were t-pinostilbenoside and t-piceid (Tables 1–4). The highest content of these compounds was observed in winter and reached 28.4 mg/g DW and 22.4 mg/g DW (Table 1), respectively.

**Table 1.** Stilbene content and repertoire in the bark of pine *Pinus koraiensis* (mg per g of the dry weight (DW)) collected in winter, spring, summer, and autumn. When using one-way analysis of variance (ANOVA) with Tukey's pairwise comparisons, the value with the same letter in each row was not significantly different. *p*-value < 0.05 was considered to be statistically significant. * The numbers corresponding to the numbers of stilbenes shown in Figure 2 are shown in parentheses.

| Bark Season/Stilbene Compound | Bark Collected in Winter, mg/g DW | Bark Collected in Spring, mg/g DW | Bark Collected in Summer, mg/g DW | Bark Collected in Autumn, mg/g DW |
|---|---|---|---|---|
| t-astringin (**1**) * | 0.143 ± 0.023 [a] | 0.102 ± 0.039 [a] | 0.084 ± 0.019 [a] | 0.039 ± 0.006 [b] |
| t-piceid (**2**) | 22.417 ± 3.251 [a] | 16.2 ± 1.587 [a,b] | 18.68 ± 2.969 [a,b] | 14.333 ± 0.813 [b] |
| cis-piceid (**3**) | 0.391 ± 0.027 [a] | 0.277 ± 0.051 [b] | 0.029 ± 0.005 [d] | 0.097 ± 0.029 [c] |
| t-isorhapontin (**4**) | 0.854 ± 0.132 [a] | 0.497 ± 0.138 [a] | 0.607 ± 0.085 [a] | 0.868 ± 0.258 [a] |
| t-pinostilbenoside (**5**) | 28.359 ± 4.272 [a] | 18.39 ± 3.071 [a,b] | 23.58 ± 4.036 [a,b] | 17.95 ± 1.658 [b] |
| t-resveratrol (**6**) | 0.868 ± 0.094 [a,b] | 0.765 ± 0.108 [b] | 0.881 ± 0.196 [a,b] | 1.487 ± 0.327 [a] |
| t-pinostilben (**7**) | 0.744 ± 0.095 [a] | 0.645 ± 0.122 [a] | 0.667 ± 0.135 [a] | 1.014 ± 0.155 [a] |
| Total | 53.777 ± 7.657 [a] | 36.89 ± 4.946 [a,b] | 44.53 ± 7.341 [a,b] | 35.79 ± 2.681 [b] |

**Table 2.** Stilbene content and repertoire in the needles of pine *Pinus koraiensis* (mg per g of the dry weight (DW)) collected in winter, spring, summer, and autumn. When using one-way analysis of variance (ANOVA) with Tukey's pairwise comparisons, the value with the same letter in each row was not significantly different. *p*-value < 0.05 was considered to be statistically significant. * The numbers corresponding to the numbers of stilbenes shown in Figure 2 are shown in parentheses.

| Needles Season/Stilbene Compound | Needles Collected in Winter, mg/g DW | Needles Collected in Spring, mg/g DW | Needles Collected in Summer, mg/g DW | Needles Collected in Autumn, mg/g DW |
|---|---|---|---|---|
| t-astringin (**1**) * | 0.012 ± 0.002 [c] | 0.026 ± 0.009 [c] | 0.065 ± 0.002 [b] | 0.148 ± 0.014 [a] |
| t-piceid (**2**) | 0.062 ± 0.007 [a] | 0.044 ± 0.008 [a] | 0.024 ± 0.002 [b] | 0.021 ± 0.005 [b] |
| cis-piceid (**3**) | 0.005 ± 0.001 [a] | 0.005 ± 0.001 [a] | 0.006 ± 0.002 [a] | 0.009 ± 0.002 [a] |
| t-isorhapontin (**4**) | 0.033 ± 0.004 [a] | 0.042 ± 0.005 [a] | 0.007 ± 0.002 [b] | 0.004 ± 0.002 [b] |
| t-pinostilbenoside (**5**) | 0.074 ± 0.009 [a] | 0.039 ± 0.007 [b] | 0.024 ± 0.005 [b] | 0.033 ± 0.007 [b] |
| t-resveratrol (**6**) | 0.007 ± 0.002 [a] | 0.011 ± 0.002 [a] | 0.002 ± 0.001 [b] | 0.016 ± 0.011 [a] |
| t-pinostilben (**7**) | 0.005 ± 0.002 [a] | 0.003 ± 0.001 [a,b] | 0 [b] | 0.005 ± 0.001 [a] |
| Total | 0.198 ± 0.123 [a] | 0.170 ± 0.016 [a,b] | 0.130 ± 0.008 [b] | 0.236 ± 0.025 [a] |

**Table 3.** Stilbene content and repertoire in the young branch of pine *Pinus koraiensis* (mg per g of the dry weight (DW)) collected in winter, spring, summer, and autumn. When using one-way analysis of variance (ANOVA) with Tukey's pairwise comparisons, the value with the same letter in each row was not significantly different. *p*-value < 0.05 was considered to be statistically significant. * The numbers corresponding to the numbers of stilbenes shown in Figure 2 are shown in parentheses.

| Branch Season/Stilbene Compound | Branch Collected in Winter, mg/g DW | Branch Collected in Spring, mg/g DW | Branch Collected in Summer, mg/g DW | Branch Collected in Autumn, mg/g DW |
|---|---|---|---|---|
| t-astringin (**1**) * | 0.010 ± 0.002 [b] | 0.023 ± 0.006 [a] | 0.029 ± 0.002 [a] | 0.031 ± 0.008 [a] |
| t-piceid (**2**) | 2.684 ± 0.557 [a] | 2.088 ± 0.058 [a,b] | 1.102 ± 0.052 [b] | 1.239 ± 0.282 [a,b] |
| cis-piceid (**3**) | 0.087 ± 0.012 [a] | 0.077 ± 0.012 [a] | 0.017 ± 0.009 [b] | 0.002 ± 0.001 [b] |
| t-isorhapontin (**4**) | 0.114 ± 0.013 [a,b] | 0.091 ± 0.007 [b] | 0.124 ± 0.012 [a] | 0.102 ± 0.007 [a,b] |
| t-pinostilbenoside (**5**) | 3.205 ± 0.527 [a] | 2.959 ± 0.527 [a] | 1.995 ± 0.151 [b] | 2.646 ± 0.251 [a,b] |
| t-resveratrol (**6**) | 0.082 ± 0.018 [a] | 0.047 ± 0.023 [a,b] | 0.004 ± 0.001 [c] | 0.009 ± 0.001 [b] |
| t-pinostilben (**7**) | 0.137 ± 0.026 [a] | 0.082 ± 0.032 [a,b] | 0.015 ± 0.002 [b] | 0.058 ± 0.005 [b] |
| Total | 6.322 ± 1.145 [a] | 5.368 ± 1.132 [a,b] | 3.288 ± 0.155 [b] | 4.088 ± 0.535 [a,b] |

**Table 4.** Stilbene content and repertoire in the wood of pine *Pinus koraiensis* (mg per g of the dry weight (DW)) collected in winter, spring, summer, and autumn. When using one-way analysis of variance (ANOVA) with Tukey's pairwise comparisons, the value with the same letter in each row was not significantly different. *p*-value < 0.05 was considered to be statistically significant. * The numbers corresponding to the numbers of stilbenes shown in Figure 2 are shown in parentheses.

| Wood Season/Stilbene Compound | Wood Collected in Winter, mg/g DW | Wood Collected in Spring, mg/g DW | Wood Collected in Summer, mg/g DW | Wood Collected in Autumn, mg/g DW |
|---|---|---|---|---|
| t-astringin (**1**) * | 0.018 ± 0.003 [b] | 0.250 ± 0.141 [a,b] | 0.462 ± 0.03 [a] | 0.010 ± 0.003 [b] |
| t-piceid (**2**) | 2.920 ± 0.23 [a] | 1.013 ± 0.072 [a,b] | 0.858 ± 0.074 [b] | 0.179 ± 0.038 [c] |
| cis-piceid (**3**) | 0.025 ± 0.005 [a] | 0.001 ± 0.001 [c] | 0 [c] | 0.008 ± 0.003 [b] |
| t-isorhapontin (**4**) | 0.073 ± 0.012 [a] | 0.016 ± 0.007 [b] | 0 [c] | 0 [c] |
| t-pinostilbenoside (**5**) | 3.116 ± 0.24 [a] | 1.470 ± 0.120 [b] | 1.335 ± 0.177 [b] | 0.279 ± 0.057 [c] |
| t-resveratrol (**6**) | 0.022 ± 0.003 [a] | 0.016 ± 0.007 [a] | 0.029 ± 0.008 [a] | 0.006 ± 0.001 [b] |
| t-pinostilben (**7**) | 0.024 ± 0.002 [a] | 0.013 ± 0.004 [a,b] | 0.031 ± 0.011 [a] | 0.006 ± 0.002 [b] |
| Total | 6.195 ± 0.483 [a] | 2.779 ± 0.049 [b] | 2.716 ± 0.295 [b] | 0.489 ± 0.096 [c] |

In turn, the needles of pine *P. koraiensis* contained significantly lower amounts of stilbenes in winter and autumn, 0.198 mg/g DW and 0.236 mg/g DW, respectively (Table 2). The lowest total content of stilbenes in needles was detected in the summer period (0.13 mg/g DW). It is interesting to note that the content of t-astringin in the needles of *P. koraiensis* collected in summer and autumn increased to 0.065 and 0.148 mg/g DW, respectively, but the content of other stilbenes remained at the same level or decreased compared to the winter and spring needle samples (Table 2). Glycosylated and methylated forms of resveratrol or piceatanol (t-astringin, t-piceid, t-isorhapontin, t-pinostilbenoside, Figure 1) constituted the bulk of all stilbenes in the needles (Table 2). However, resveratrol content did not exceed 0.02 mg/g DW (Table 2), and piceatannol was not identified due to low levels (less than 0.001 mg/g DW).

The total content of stilbenes in young branches of pine *P. koraiensis* reached 6.3 mg/g DW in winter samples, which gradually decreased to 3.3 mg/g DW in summer samples (Table 3). The predominant stilbenes in all samples of *P. koraiensis* young branches were t-piceid and t-pinostilbenoside, with their maximum contents reaching 2.7 mg/g DW and 3.2 mg/g DW, respectively (Table 3). The content of individual stilbenes decreased or remained at the same level in summer samples of needles, with the exception of t-astringin, whose content increased by three times in comparison with the needles collected in the winter.

The total content of stilbenes in pine *P. koraiensis* wood was significantly higher in winter, up to 6.2 mg/g DW, which significantly decreased to a level of 0.5 mg/g DW in the autumn samples (Table 4). As in the case of needles, bark, and branches, the main stilbenes were glycosylated forms of stilbenes, namely, t-piceid and t-pinostilbenoside: more than 95% of all stilbenes (Table 4).

We also analyzed the total content and composition of stilbenes in various parts of *P. koraiensis* strobiles (scales, seed shells, and pine nuts) collected in autumn (Table 5). The interest to the pine strobiles is explained by the fact that the pine seeds are actively used in food industry and can be a source of valuable stilbenes in human nutrition. This study showed that the content and diversity of stilbenes in scales, seed shells, and pine nuts of pine was lower compared to the vegetative parts of the plant (Table 5). In the scales, we detected five individual stilbenes (t-astringin, t-piceid, t-pinostilbenoside, t-resveratrol, t-pinostilben), the total contents of which did not exceed 0.13 mg/g DW (Table 5). The total content of stilbenes in the seed shells was no more than 0.05 mg/g DW, and, in nuts, stilbenes were completely absent (Table 5). Thus, nuts (the edible part) cannot be a source of stilbenes in human nutrition.

**Table 5.** Stilbene content and repertoire in the various strobile parts (seed scales, shell of seeds, and seeds) of pine *Pinus koraiensis* (mg per g of the dry weight (DW)) collected in autumn. When using one-way analysis of variance (ANOVA) with Tukey's pairwise comparisons, the value with the same letter in each row was not significantly different. *p*-value < 0.05 was considered to be statistically significant. * The numbers corresponding to the numbers of stilbenes shown in Figure 2 are shown in parentheses.

| Strobiles Parts/Stilbene | Scales, mg/g DW | Seed Shells, mg/g DW | Nuts, mg/g DW |
|---|---|---|---|
| t-astringin (**1**) * | 0.017 ± 0.003 [a] | 0.007 ± 0.001 [b] | 0 |
| t-piceid (**2**) | 0.036 ± 0.009 [a] | 0.035 ± 0.014 [a] | 0 |
| cis-piceid (**3**) | 0 | 0 | 0 |
| t-isorhapontin (**4**) | 0 | 0 | 0 |
| t-pinostilbenoside (**5**) | 0.051 ± 0.014 [a] | 0.008 ± 0.005 [b] | 0 |
| t-resveratrol (**6**) | 0.004 ± 0.002 | 0 | 0 |
| t-pinostilben (**7**) | 0.005 ± 0.001 | 0 | 0 |
| Total | 0.127 ± 0.031 [a] | 0.050 ± 0.013 [b] | 0 |

### 3.2. Expression Analysis of PkSTS Genes

Based on the STS sequences from related plant species, *Pinus densiflora*, *Pinus strobus*, and *Pinus massoniana* available in GenBank, primers were selected to the beginning and the end of the protein-coding part of the *STS* gene. The cDNA fragments of interest were amplified, cloned, and sequenced. Five independent transformations were performed. Using a high-fidelity polymerase (Tersus, Eurogen), 20 full-length *PkSTS* gene sequences from the pine were cloned, all of which were highly homologous to the previously described *PsSTS*. Based on the nucleotide sequences, we isolated eight different transcripts: *PkSTS1a*, *1b*, *2a*, *2b*, *2c*, *3a*, *3b*, *3c* (Supplementary Figure S1). The most frequent was *PkSTS1a*, 11 clones, 55% of all analyzed clones. *PkSTS1b* and *PkSTS3a* were repeated two times each, and the rest was detected only one time each (*PkSTS2a, 2b, 2c, 3b, 3c*).

The expression of all *PkSTSs* strongly depended on the season (Figure 3). The expression of *PkSTS1* changed to a greater extent, with the highest level of expression detected in winter, and dramatically decreased in summer (Figure 3a). The expression of *PkSTS2* and *PkSTS3* changed to a lesser extent compared to *PkSTS1*, with the highest level of *PkSTS2* and *PkSTS3* detected in winter (Figure 3b,c). Based on the expression data, as well as on the content of stilbenes in needles of *P. koraiensis* in different seasons, it can be assumed that *PkSTS1* played the main role in the biosynthesis of stilbenes, but this assumption requires additional studies. Overall, we described the presence of three *STS* genes expressed in the needles of *P. koraiensis*, whereas the presence of only one *STS* gene, the product of which was pinosylvin, was previously described for *P. strobus* [24]. *P. densiflora* has also been shown to contain three *STS* genes that also form pinosylvin [47]. However, we have shown that the content of pinosylvin and its derivatives in the tissues of *P. koraiensis* was not considerable. In *P. koraiensis* samples, the main stilbenes were methylated and glycosylated forms of resveratrol. It is possible that the protein products of the newly described *STS* genes could synthesize resveratrol, which is further modified into other stilbenes, but this assumption requires further investigation.

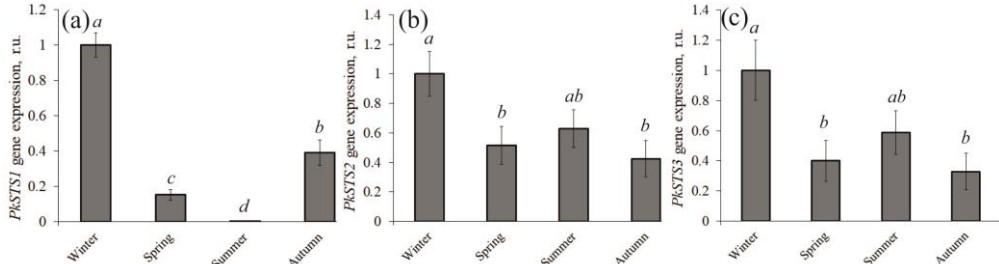

**Figure 3.** The expression of *PkSTS* genes ((**a**)—*PkSTS1*; (**b**)—*PkSTS2*; (**c**)—*PkSTS3*) in the *Pinus koraiensis* that have been collected in the winter, spring, summer, and autumn. The means in each column followed by the same letter did not differ when using one-way analysis of variance (ANOVA) with Tukey's pairwise comparisons.

## 4. Discussion

The wood of coniferous trees, as well as waste from its processing in the paper and woodworking industries (such as knots, bark, and branches), are considered as a good source of valuable polyphenols, including stilbenes [48]. The content of stilbenes in the bark of *Picea abies* and *Picea jezoensis* has been shown to reach 120 mg/g DW and 165 mg/g DW, respectively [27,31]. It is important to note that the prevalent stilbenes, in this case, were t-astringin and t-isorhapontin. Recently, Gabaston et al. [49] not only characterized *Pinus pinaster* stilbenes using UHPLC-DAD-MS and NMR but also studied their antifungal activity against grapevine pathogens.

To the best of our knowledge, this work firstly presented a detailed qualitative and quantitative analysis of stilbenes in *P. koraiensis* and described the *STS* genes of *P. koraiensis* responsible for stilbene biosynthesis. Previous studies and data obtained in this work showed that conifers of the Pinaceae family are rich in rare compounds, including stilbenes.

Stilbenes consist of two aromatic rings linked by a C2 ethylene side chain, formed by the cyclization of 3-malonyl-CoA with cinnamoyl-CoA, coumaroyl-CoA, or caffeyl-CoA, followed by hydroxylation of the aromatic rings [35].

At the moment, a fairly large number of plant species is known to synthesize and accumulate various stilbenes, which can exist in the form of two stereoisomers: *trans*-stilbene (t-stilbene, (E)-stilbene) and *cis*-stilbene ((Z)-stilbene) [50]. The double bond determines the *cis-trans* isomerism of the stilbene molecule. The *cis*-isoform is less stable due to steric hindrance caused by the proximity of the phenyl groups, which determines a significant difference in the properties of the isomers. Under the influence of lightning, isomers are able to convert into each other [51].

Typically, plants synthesize t-resveratrol as a monomeric stilbene, which undergoes further modifications such as methylation, glycosylation, and oxidation, resulting in a wide variety of other stilbenes. It is important to note that some studies describe the enzyme pinosylvin synthase (EC 2.3.1.146), e.g., in *P. sylvestris* [52], *P. strobes* [53], and *P. densiflora* [47], which is involved in the formation of pinosylvin, a different monomeric stilbene. It should be noted that STS enzymes can accept different cinnamic acid derivatives as substrates, and a single enzyme can be responsible for the biosynthesis of different stilbenes, depending on the parent molecule [37,52,53]. While this study did not detect pinosylvin in the pine *P. koraiensis*, it showed the presence of t-resveratrol and its glycoside t-piceid. These results suggest that the synthesis of stilbenes in *P. koraiensis* proceeds through the formation of t-resveratrol as a monomeric stilbene with its subsequent modifications, leading to the formation of t-pinostilbenoside, the predominant stilbene in all parts of *P. koraiensis*. T-pinostilbenoside is a methylated t-resveratrol glucoside. At present, little is known about the biological functions of t-pinostilbenoside. However, it is known that methylated resveratrol analogues exhibit similar biological activities comparable to that of resveratrol. Methylated resveratrol analogues are known to have better bioavailability because they are easier transported into the cell and are more resistant to degradation [54]. It was also shown that glycosylated forms of stilbenes are better soluble in water [55], which facilitates their transport through the vascular system of plants and the synthesis zone (needles) to the accumulation zone (bark).

In most stilbene-producing plant species, stilbenes accumulate in small amounts, i.e., less than 10 mg/g DW. However, a number of plant species are capable of accumulating stilbenes in amounts of more than 10 mg/g DW. It is interesting to note that these species belong to four plant families, namely, Pinaceae, Moraceae, Polygonaceae, and Vitaceae. To the best of our knowledge, the highest content of stilbenes reaching 120–160 mg/g DW was detected in the bark of *P. abies* and *P. jezoensis* [27,31]: 54 mg/g—in the root bark of *Morus albus* DW [56]; 19 mg/g DW—in the roots of *Polygonum cuspidatum* [9]; and 10 mg/g DW—in the roots of *Vitis amurensis* [57].

In this work, the highest total content of stilbenes was detected in the winter pine *P. koraiensis* (up to 53 mg/g DW in the bark), with the exception of needles, where the maximum number of stilbenes was in the autumn samples. During the period of active plant growth (spring and summer), the number of stilbenes in the bark, needles, and branches decreased by 1.2–1.9 times. Our results confirmed the previously obtained data on seasonal fluctuations on the level of stilbenes in the needles and bark of *P. jezoensis*, indicating a significant decrease in the level of stilbenes in *P. jezoensis* material collected in summer [31,35]. The detected decrease in the total content of stilbenes during the year was associated with a decrease in the content of two major stilbenes, t-piceid and t-pinostilbenoside. At the same time, the content of t-astringin increased in summer and autumn in most cases. The content of stilbenes in scales was comparable to stilbene content in the summer samples of needles and did not exceed 0.13 mg/g DW. It is known that stilbene biosynthesis is activated in response to UV irradiation, drought, salinity, pathogens, and other environmental stresses [58]. Therefore, it is possible that the seasonal variability in stilbene content contributed to the protection of spruce tissues from adverse environmental conditions.

## 5. Conclusions

In conclusion, the data obtained indicated that *P. koraiensis* could be a rich source of stilbenes, especially when using technologies for a deeper complete processing of wood waste. Winter is the most suitable season to obtain stilbenes from *P. koraiensis* as the year season with the highest stilbene contents in the pine. The data also revealed that the winter bark of the pine was one of the richest sources of stilbenes, especially considering t-piceid and t-pinostilbenoside.

**Supplementary Materials:** The following are available online at https://www.mdpi.com/article/10.3390/f14061239/s1. Table S1. List of the stilbene derivatives identified in the methanol extracts of pine *Pinus koraiensis* bark, needles, branches, wood, and strobiles. Table S2. Stilbene content in different Pinaceae species. Table S3. Primers used for amplification of *Pinus koraiensis* cDNAs in PCR. Table S4. Comparison of nucleotide and deduced amino acid sequences of the STS genes from *Pinus strobus* and *Pinus koraiensis*. Figure S1. Nucleotide sequence of the *PkSTS* genes and primes used for real-time PCR.

**Author Contributions:** K.V.K. and A.S.D. performed research design, data analysis, interpretation, and paper preparation. A.R.S. and V.P.G. performed collection of plant material, extraction, and HPLC-MS analysis. All authors have read and agreed to the published version of the manuscript.

**Funding:** This work was supported by the grant #22-16-00078 from the Russian Science Foundation, https://rscf.ru/en/project/22-16-00078 (accessed on 5 June 2023).

**Data Availability Statement:** Data are contained within the article and Supplementary Materials.

**Conflicts of Interest:** We declare that we have no conflict of interest.

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
