# Peer review of "Stilbene Content and Expression of Stilbene Synthase Genes in Korean Pine Pinus koraiensis Siebold & Zucc"

_forests, doi:10.3390/f14061239_

Round 1
Reviewer 1 Report
Dear authors I think your article is of scientific interest for readers and describes new important results in stilbene content in different seasons in several parts of Korean pine. I have worked on your text and I made some corrections detailed in the notes. I suggest to add the number of your Voucher Specimen.
A herbarium specimen voucher is usually a pressed, mounted plant sample with collection data deposited for future reference. It supports research work and may be examined to verify the identity or study other elements of the specimen. A voucher specimen must be deposited in a recognized herbarium committed to long-term maintenance.
I also suggest to rename some words used to describe the different parts of the pine strobiles to avoid misunderstanding: scales, seed shells and pine nuts (the edible part),and to substitute them in all the text.
The description of Table 4 is missing.
The References numbers should be added in Table S1
In row 62 I think you are referring to Table S2 instead of Table S1
In Table S2 the Reference numbers in the fifth column should have the same number of the articles reported in the References at the end of the text

I suggest minor editing in English language
Author Response
- I suggest to add the number of your Voucher Specimen. A herbarium specimen voucher is usually a pressed, mounted plant sample with collection data deposited for future reference. It supports research work and may be examined to verify the identity or study other elements of the specimen. A voucher specimen must be deposited in a recognized herbarium committed to long-term maintenance.
Answer: Thank you for this recommendation. The exact determination of the plant species was carried out at the laboratory of botany of the Federal Scientific Center for Biodiversity by the deeply respected Prof. Andrey A. Goncharov (Scopus ID 6603752347). We do not doubt his professionalism on this matter. Unfortunately, at the moment, we do not have the opportunity to prepare a herbarium specimen for deposition and storage in a recognized herbarium, but in future work we will definitely take into account this remark.
- I also suggest to rename some words used to describe the different parts of the pine strobiles to avoid misunderstanding: scales, seed shells and pine nuts (the edible part), and to substitute them in all the text.
Answer: Thank you for this suggestion. We agree and we have made the necessary changes in accordance with your suggestions.
- The description of Table 4 is missing.
Answer: Thank you. Corrected.
- The References numbers should be added in Table S1
Answer: Thank you for the comment. The data presented in Table S1 were obtained in this work. We obtained them on an instrument during the analysis of standards in the present work.
- In row 62 I think you are referring to Table S2 instead of Table S1.
Answer: Thanks for this comment, you're right. Corrected.
- In Table S2 the Reference numbers in the fifth column should have the same number of the articles reported in the References at the end of the text.
Answer: Thank you. Corrected.
Reviewer 2 Report
Stilbenes constitute a significant group of plant phenolic compounds known for their antioxidant, immunomodulatory, anti-inflammatory, and anti-angiogenic effects, among others. This study focuses on the detection of seven derivative components of stilbenes using HPLC-PDAD-MS. The conditions for high-performance liquid chromatography were optimized, resulting in well-separated peaks of the target compounds. Chromatogram accumulation results revealed significant variations in contents across different seasons and tissues. Additionally, three stilbene synthase genes, namely PkSTS1, PkSTS2, and PkSTS3, were successfully cloned and analyzed for their expression patterns in plants using real-time PCR. A positive correlation was observed between the expression levels of these genes and the accumulation of metabolites. Throughout the paper, the experimental details are provided in a relatively comprehensive and detailed manner, without grammatical errors. The findings of this research hold significant scientific significance for understanding the biosynthesis of stilbene compounds.
Author Response
Answer: Thank you for your kind feedback on this work.
Reviewer 3 Report
1. In order to make readers better understand the research content, the author should consider drawing a Figure of stilbene biosynthesis pathway.
2. Obvious typo in line 181.
3. Try to use Figures instead of Tables to display stilbene content.
4. The research methods need to be described carefully, rather than "as described before".
5. Use amino acid sequence alignment to show sequence differences between PkSTSs.
6. Which PkSTS may assume the main function of Stilbene biosynthesis?
7. The author needs to consider whether to draw a model diagram to show the main conclusions of this paper: The high expression of STS gene in winter leads to the accumulation of stilbenes.
8. Glycosylated and methylated forms of stilbenes are the main forms of accumulation. What is their biological significance?
9. Stilbene is highest in winter. Does it make the plant more resistant to freezing?
Author Response
- In order to make readers better understand the research content, the author should consider drawing a Figure of stilbene biosynthesis pathway.
Answer: Thank you for this recommendation. This paper describes the pathway for the biosynthesis of stilbenes, which gives a sufficient understanding of the process. Stilbene biosynthesis schemes are presented in many works, including references (Kiselev et al. 2016, Tyunin et al. 2018, Hammerbacher et al. 2011, Schmidlin et al. 2008, Line 50-62)
- Obvious typo in line 181.
Answer: Corrected.
- Try to use Figures instead of Tables to display stilbene content.
Answer: Thank you very much for your suggestion. The option of providing data on the content of stilbene in the form of a picture was considered by us when writing the article. However, the presentation in the form of tables seemed to us more informative and easy to understand, so we decided to leave the tables in the new version of the manuscript.
- The research methods need to be described carefully, rather than "as described before".
Answer: Thank you for your comment. The methods have been described in more detail.
- Use amino acid sequence alignment to show sequence differences between PkSTSs.
Answer: All obtained amino acid and nucleotide sequences of STS demonstrated a high percentage of identity between each other and with the previously described STS from other species of the genus Pinus (97–98%, Identities), confirming the attribution of the obtained sequences to the stilbene synthase family. This information has been added to the text of the manuscript. Please, see Line 279-282 and table S4.
- Which PkSTS may assume the main function of Stilbene biosynthesis?
Answer: Based on the expression data, as well as on the content of stilbenes in needles of P. koraiensis in different seasons, it can be assumed that PkSTS1 plays the main role in the biosynthesis of stilbenes, but this assumption requires additional studies.
This hypothesis has been added to the text of the manuscript. Please, see Line 291-293.
- The author needs to consider whether to draw a model diagram to show the main conclusions of this paper: The high expression of STS gene in winter leads to the accumulation of stilbenes.
Answer: Thank you for this recommendation. We added a graphical abstract when we submitted this time the revised manuscript to emphasize the main idea and results of this work. We also added it to the Supplementary material (see Graphical abstract).
- Glycosylated and methylated forms of stilbenes are the main forms of accumulation. What is their biological significance?
Answer: Thank you for this interesting question. It is known that methylated resveratrol analogues exhibit valuable biological activities comparable to that of resveratrol. Furthermore, methylated resveratrol analogues are known to have better bioavailability because they are easier transported into the cell and are more resistant to degradation (Kang et al. 2014). Also, it has been shown that glycosylated forms of stilbenes are better soluble in water (Shimoda et al. 2020), which facilitates their transport through the vascular system of plants and the synthesis zone (needles) to the accumulation zone (bark).
This information has been added to the text of the manuscript. Please, see Line 343-345.
- Stilbene is highest in winter. Does it make the plant more resistant to freezing?
Answer: Indeed, this is an interesting and important question. The effect of stilbenes on plant cold tolerance was not investigated in this work, but it has recently been reported that stilbenes (resveratrol, piceid) did not confer cold tolerance on VaSTS transgenic Arabidopsis plants (Ogneva et al. 2021). Moreover, our new results show that exogenous application of stilbenes (resveratrol, piceid) to Arabidopsis plants did not improve plant cold tolerance. This suggests that the accumulation of stilbene in Pine does not contribute to its resistance to low temperatures, but this assumption requires a separate study.